# Genome-Wide Association Analysis Identified Variants Associated with Body Measurement and Reproduction Traits in Shaziling Pigs

**DOI:** 10.3390/genes14020522

**Published:** 2023-02-18

**Authors:** Qun Lan, Qiuchun Deng, Shijin Qi, Yuebo Zhang, Zhi Li, Shishu Yin, Yulian Li, Hong Tan, Maisheng Wu, Yulong Yin, Jun He, Mei Liu

**Affiliations:** 1College of Animal Science and Technology, Hunan Agricultural University, Changsha 410128, China; 2Xiang Dong Experiment Station, Hunan Provincial Pig Industrial Technology System, Xiangtan 411100, China; 3Key Laboratory of Agro-Ecological Processes in Subtropical Region, Institute of Subtropical Agriculture, Chinese Academy of Sciences; Hunan Provincial Engineering Research Center for Healthy Livestock and Poultry Production; Scientific Observing and Experimental Station of Animal Nutrition and Feed Science in South-Central, Ministry of Agriculture, Changsha 410125, China; 4Kunpeng Institute of Modern Agriculture at Foshan, Foshan 528226, China

**Keywords:** Shaziling pigs, SNP, body measurement trait, reproductive trait, GWAS

## Abstract

With the increasing popularity of genomic sequencing, breeders pay more attention to identifying the crucial molecular markers and quantitative trait loci for improving the body size and reproduction traits that could affect the production efficiency of pig-breeding enterprises. Nevertheless, for the Shaziling pig, a well-known indigenous breed in China, the relationship between phenotypes and their corresponding genetic architecture remains largely unknown. Herein, in the Shaziling population, a total of 190 samples were genotyped using the Geneseek Porcine 50K SNP Chip, obtaining 41857 SNPs for further analysis. For phenotypes, two body measurement traits and four reproduction traits in the first parity from the 190 Shaziling sows were measured and recorded, respectively. Subsequently, a genome-wide association study (GWAS) between the SNPs and the six phenotypes was performed. The correlation between body size and reproduction phenotypes was not statistically significant. A total of 31 SNPs were found to be associated with body length (BL), chest circumference (CC), number of healthy births (NHB), and number of stillborns (NSB). Gene annotation for those candidate SNPs identified 18 functional genes, such as *GLP1R*, *NFYA*, *NANOG*, *COX7A2*, *BMPR1B*, *FOXP1*, *SLC29A1*, *CNTNAP4*, and *KIT*, which exert important roles in skeletal morphogenesis, chondrogenesis, obesity, and embryonic and fetal development. These findings are helpful to better understand the genetic mechanism for body size and reproduction phenotypes, while the phenotype-associated SNPs could be used as the molecular markers for the pig breeding programs.

## 1. Introduction

Domesticated pigs account for a high proportion of meat production in the world [1]. In the pig industry, reproductive performance is directly correlated with production efficiency and economic profits. The reproductive traits, especially litter traits, exhibited low heritability (less than 0.1) and were regulated by many genes and their interactions [2]. Traditional breeding programs for genetic improvement with low heritability, complex traits often progress slowly, which has undoubtedly elevated the need for molecular breeding [3]. Genomic selection (GS) and Molecular marker-assisted selection (MAS) have been demonstrated to have remarkable superiority in improving traits with low heritability [4]. Obtaining more accurate genetic information, such as quantitative trait loci (QTLs), single-nucleotide polymorphisms (SNPs), and candidate genes, related to reproductive traits will greatly increase the selection efficiency of GS and MAS.

From the perspective of breeders, phenotypic records for body measurement traits are vital for evaluating the production performance and making a future breeding schedule. The body measurement traits present low-to-moderate heritability [5,6,7,8]. The study revealed that several body measurement traits could be regarded as indicator traits for improving production traits [9]. Importantly, some recent studies pointed out the vital relationship between the body measurement traits and reproductive performance [10,11], hinting that the body conformation traits of female animals might be important for improving reproduction performance.

The last decade has witnessed the application of genome-wide association studies (GWAS) in identifying the genetic markers covering the whole genome for diverse body measurement and reproduction phenotypes. According to the latest data from PigQTLdb (https://www.animalgenome.org/cgi-bin/QTLdb/SS/index, accessed on 9 January 2023), there are 35,846 QTLs related with 693 different traits in pigs that have been mapped by 773 publications. Among the 1281 QTLs associated with litter traits, there were only 466 QTLs related to total numbers born (TNB), 374 QTLs related to number of healthy births (NHB), 138 QTLs related to numbers of stillborn (NSB), and 149 QTLs related to the number of mummified pigs (MUMM). Regarding the body measurement traits, 188 QTLs related to chest circumference (CC), and 219 QTLs related to body length (BL). The Shaziling pig is primarily raised in the Hunan Province of China and is characterized by low growth rate, good meat quality, and strong resistance to diseases [12]. Earlier genomics studies for Shaziling pigs were primarily focused on the genetic diversity [13], skeletal muscle growth [14], and umbilical hernia [15]. However, so far, identifying key molecular markers through the GWAS technique for improving production performance is still progressing slowly.

In this study, we aim to uncover the relationship between body measurement traits and reproduction phenotypes, and to clarify their associated genetic variations in Shaziling pigs. To this end, the body measurement traits (BL and CC) and reproductive traits (TNB, NHB, MUMM, and NSB) were measured and collected, respectively. Subsequently, based on the Geneseek Porcine 50K SNP Chip data, the SNPs were detected in those Shaziling pigs. The GWAS was conducted to identify potentially critical SNPs and QTL regions. This work paves the way for the future genetic improvement of pigs with efficient growth and reproduction performances.

## 2. Materials and Methods

### 2.1. Animals and Phenotypes

Our studied population (Shaziling pigs) were raised on the conservation farm (Xiangtan, Hunan, China). Animals were subjected to the same nutritional and management conditions. Firstly, the BL and CC of 190 sows (from 240 to 265 days) were measured according to the method applied in previous research [7]. Briefly, these two body size traits were measured by a tape or a meter ruler. Then, the same batch of 190 sows in the production stage was sampled and litter records were collected in the first parity. TNB, NHB, MUMM, and NSB were recorded for each parity. Finally, these six traits were used for the following analysis. The ethics committee of Hunan Agricultural University approved all the experimental procedures in this study (Permit Number: 20210701).

### 2.2. Animals and Genotypes

The ear tissues of 190 Shaziling pigs were collected for genomic DNA extraction according the protocol of Accurate SteadyPure Universal Genomic DNA Extraction Kit (Accurate Biology, Changsha, China). The concentration and quality of genomic DNA were testing by Nanodrop 2000 (Thermo Scientific, Waltham, MA, USA) and 1.5% agarose gel electrophoresis, respectively. The final qualified DNA samples were genotyped using Geneseek Porcine 50K SNP Chip (GeneSeek, Lincoln, NE, USA).

### 2.3. Data Quality Control

For phenotypes, the R (version 4.2.1) was used to perform descriptive statistical analyses to check the data quality including the mean, standard deviation, coefficient of variance, minimum and maximum values for each trait. The plot of normal distribution and correlation coefficients among phenotypic values were all visualized and calculated by the “GGally” package in R. For the genotypes, PLINK software (version 1.90 beta) was used to conduct quality control to achieve high-quality common SNPs [16]. Briefly, the filtered procedures were performed as follow: (1) individual call rate >90%; (2) minor allele frequency (MAF) >5%; and (3) *p*-value > 10^−6^ for the Hardy-Weinberg equilibrium (HWE) test. Furthermore, we excluded the SNPs located on the sex chromosome and unmatched regions.

### 2.4. SNP-Based Heritability

The heritability estimates of phenotypes were calculated using a restricted genomic-relatedness-based restricted maximum-likelihood (GREML) analysis, as implemented in GCTA software (version 1.94) [17]. Briefly, based on the high-quality SNPs, the genetic relationship matrix (GRM) was used to estimate the genetic relationships between pairwise individuals from all of the autosomal SNPs, and then the GRM and phenotypes were included in the restricted maximum likelihood (REML) analysis to estimate the variance explained by the SNPs.

### 2.5. Population Structure and Kinship Analyses

Principal component analysis (PCA) was performed to identify the population structure of Shaziling pigs to determine whether principal components (PCs) should be included in the GWAS model. Simply, the eigenvalues and eigenvectors were calculated by GCTA software based on high quality filtered SNPs. The lollipop-plot of the top ten principal components were visualized using the “ggplot2” package in R. A heat map was plotted against the kinship matrix to exhibit the level of relatedness among all individuals within the Shaziling population.

### 2.6. GWAS

The association analysis between each SNP marker and phenotypic data was performed using the mixed linear model in the GEMMA software [18]. The model described was as follows:***y*** = ***W**α*** + ***X**β*** + ***μ***+ ***e***(1)
where ***y*** represents the vector of phenotypes (BL, CC, TNB, NHB, NSB, and MUMM) for each individual; ***W**α*** included the population structure effect and fixed effects, briefly, age (in days) was used as fixed effects for BL and CC, and the individual of boar used for mating was used as fixed effects for TNB, NHB, NSB, and MUMM; ***X**β*** is the marker effect to be tested; ***μ****~N* (0, *K*ϕ^2^) represents the polygenic effect; ***e****~N* (0, *Iσ*^2^) refers to the residual effect, and *K* is the kinship matrix generated from the SNPs. Similar to previous GWAS research [19,20], the Bonferroni-adjusted genome-wide significant threshold was set as 0.05/N, and the suggestive threshold was set as 1/N, N being the total number of filtered SNPs.

### 2.7. Candidate Gene Search and Functional Annotation

To identify the candidate genes of each significant SNP, we searched for annotated genes according to their physical positions (within 500 Kb upstream or downstream) on the pig reference genome (Sscrofa 11.1) (http://asia.ensembl.org/biomart/martview/, accessed on 18 October 2022). Gene Ontology (GO) functional annotation analysis and Kyoto Encyclopedia of Genes and Genomes (KEGG) enrichment analysis were performed in the enrichment module of KOBAS 3.0 [21], and then the significant threshold for the GO terms (biological process) and KEGG pathways were set at Benjamini-Hochberg adjusted *p*-value < 0.05.

## 3. Results

### 3.1. Description of Phenotypes and Genotypes

As shown in Figure 1, phenotypes, including BL, CC, TNB, and NHB, of 190 Shaziling pigs displayed approximately normal distributions in the current study. TNB and NHB were strongly and positively correlated (r = 0.826, *p* < 0.001). BL and CC were weakly correlated (r = 0.294, *p* < 0.001). However, there was no correlation between body measurement traits (BL and CC) and reproduction traits (TNB and NHB). Descriptive statistics for the traits of BL, CC, TNB, NHB, MUMM, and NSB analyzed in the current study are shown in Table 1. In the present study, the heritability of BL and CC was 0.15 and 0.14, respectively. Except for MUMM, the other three reproduction traits were less than 0.01. The coefficient of variation (CV) ranges from 7.96% (lowest one: BL) to 9.03% (highest one: CC).

In the Shaziling population, a total of 190 samples were genotyped using the Geneseek Porcine 50K SNP Chip. A total of 41,857 SNPs were obtained for further marker analysis after a series of quality filter procedures. The distribution of the SNP information on eighteen autosome chromosomes was presented in Figure 2A. According to the SNP numbers for GWAS analysis, the genome-wide significant and suggestive threshold of the Shaziling population were 1.19 × 10^−6^ (0.05/41,857) and 2.38 × 10^−5^ (1/41,857), respectively.

### 3.2. Results of Population Structure and Kinship Analyses

The principal component analysis was performed for the tested Shaziling individuals to avoid the false-positive results caused by population stratification, which was considered as an important influencing factor to the reliability of the GWAS result. The results showed that the first four principal components could be included in the GWAS model as covariates (Figure 2B, Appendix A). Besides, a kinship matrix was also used as the covariate in the fixed effects model for GWAS analysis. The kinship heat map was shown in Appendix A.

### 3.3. GWAS Results

The GWAS results are displayed in Figure 3 and Table 2. In total, 8 SNPs surpassed the genome-wide significance level and 23 SNPs reached the suggestive threshold. In detail, on chromosome 7, the *GLP1R* gene corresponding to the significant SNP (ALGA0040227) was significantly associated with BL. Moreover, at the suggestive level, another 4 SNPs located on chromosome 7 were also associated with BL, including ALGA0039856, ASGA0032589, ALGA0040238, and INRA0024788, which were respectively mapped to genes *SNRPC*, *NFYA*, *GLP1R*, and *GLP1R*. Regarding the CC trait, 4 SNPs reached the significant level. Among these 4 SNPs, H3GA0003059 and ALGA0004562 on chromosome 1 were mapped to genes *NANOG* and *COX7A2*, respectively. ASGA0050356 and ALGA0065112 located on chromosomes 11 and 12 were respectively mapped to genes *FAM216B* and *CACNG1*. Furthermore, 9 SNPs (ASGA0057447, ALGA0111294, MARC0074335, MARC0090402, ASGA0073620, ALGA0024545, DRGA0000770, WU_10.2_12_884330, and WU_10.2_12_890035) at the suggestive-level located on chromosome 13, 8, 12, 1, 16, 4, 1, 12, and 12, respectively, were also detected to be associated with trait CC, and the genes closest to the 9 SNPs were *FOXP1*, *BMPR1B*, *DNAH9*, *ENSSSCG00000060802*, *CCNG1*, *COX6C*, *RIMS1*, *CCDC57*, and *CCDC57*, respectively. 

For reproduction traits, the *SMG1* gene on chromosome 3 was found to be nearest to suggestive SNPs (CASI0010189) related to TNB. In addition, 3 SNPs (ASGA0038253, ALGA0108557, and ALGA0040524) in significant-level located on chromosome 8, 1, and 7, respectively, were detected to be related with trait NSB, and the genes nearest to the 3 SNPs were *ARAP2*, *GTF2H5*, and *SLC29A1*, respectively. Moreover, 9 SNPs (WU_10.2_4_11278211, CASI0009401, H3GA0024817, DIAS0000597, WU_10.2_18_23986398, DBMA0000241, H3GA0048552, WU_10.2_4_11747027, and WU_10.2_7_7925995) in suggestive-level located on chromosome 4, 6, 8, 3, 18, 7, 17, 4, and 7, respectively, were also detected to be related with the NSB trait, and the genes nearest to the 8 SNPs were *FAM49B*, *CNTNAP4*, *KIT*, *POR*, *SPAM1*, *TFAP2B*, *BPIFA3*, *ENSSSCG00000060440*, and *NEDD9*. However, no suggestive or significant SNPs were found to relate to MUMM and NHB (Appendix A).

### 3.4. Enrichment Analysis

In order to obtain more insights into the functions of the 18 and 13 SNPs respectively related to the indicators of the body measurement traits (BL and CC) and reproductive phenotypes (NHB and NSB) of the Shaziling population, the corresponding candidate gene was used for enrichment analysis. The significant GO terms mainly included “GO:0008528: G protein-coupled peptide receptor activity”, “GO:0002063: chondrocyte development”, “GO:0006687: glycosphingolipid metabolic process”, “GO:0043588: skin development”, and “GO:0005246: calcium channel regulator activity” (Figure 4A). On the other hand, the KEGG pathways are mainly related to signaling pathways regulating the pluripotency of stem cells, Glycosaminoglycan degradation, and MAPK signaling pathway (Figure 4B). Other detailed annotated information can be found in Appendix A.

## 4. Discussion

To our knowledge, this is the first study evaluating the relationship between body measurement and reproductive phenotypes for female Shaziling pigs, and their associated genetic architecture by GWAS analysis. 

Previous studies have indicated that body measurement traits could be the indicator of reproduction potential in sows [22,23,24] and cows [25]. However, in the current study, we did not observe significant effects of BL and CC on reproduction traits. Similarly, previous studies in Danish crossbred sows (Yorkshire × Landrace) and the Duroc population showed no statistically significant associations between BL and reproduction traits like litter size [26,27]. These results implied that body measurement traits have little influence on reproduction performance. However, it is necessary to enlarge the Shaziling population and types of body measurement traits in future correlation analyses to validate our findings. On the other hand, studies verified that the BL and CC exhibited low to moderate heritability [28,29]. We have presented here the same tendency for heritability in the two traits, suggesting that it is practicable to improve the body dimensions in Shaziling pigs through genetic selection. In addition, the genomic heritability of reproduction traits (TNB, NHB, NSB, and MUMM) were low. As a typical quantitative trait, litter traits are co-regulated by the interactions of many genes [30]. The low heritability of reproduction traits implies that accurately identifying more candidate genes related to reproduction traits would be helpful for breeders to improve reproduction performance.

Body measurement traits such as BL and CC are usually associated with growth traits and meat production. For example, it has been reported that BL and CC were positively associated with body weight [31]. BL is a vital factor affecting livestock slaughter performance, as a longer carcass signifies longer loins and more meat production [32]. In addition, the swine carcass length was positively correlated with the total lean meat content [33]. In the current work, a series of candidate genes corresponding to different SNPs for BL and CC were identified. As for BL, one of the significant candidate genes was *GLP1R*, which was the nearest gene for three different SNPs on chromosome 7. According to a previous study, *GLP1R* was expressed in primary osteoclasts, bone marrow cells, and osteoblasts [34]. In addition, the study revealed that mice lacking *GLP1R* displayed negative effects on bone strength and quality, the apparent symptoms including impaired mechanical properties, a significant decrease in the bone’s outer diameter and cortical thickness [35]. Our GO functional annotation analysis revealed that *GLP1R* is related to the G protein-coupled peptide receptor activity. Interestingly, the G-protein coupled receptor family has been reported to correspond with regulating limb patterning and skeletal morphogenesis, as well as morphogenesis during embryonic development [36,37]. This indicated that *GLP1R* might be the critical regulator of skeletal bone mass, so it was supposed to be a strong candidate gene for BL. Except for *GLP1R*, we found that the candidate gene *NFYA* was also associated with BL. It has been demonstrated that *NFYA* was an essential modulator in governing embryonic cartilage growth [38]. Moreover, the overexpression of two *NFYA* isoforms exerts different functions on myoblasts, with *NFYAs* increasing cell proliferation and *NFYAl* promoting differentiation [39]. We contend that *NFYA* might play important roles in modulating the formation of myoblasts and cartilage that ultimately affect the development of BL. Regarding the CC, the candidate genes were *NANOG*, *COX7A2*, *CACNG1*, *FOXP1*, *BMPR1B*, *DNAH9*, *CCNG1*, *COX6C*, *RIMS1*, and *CCDC57*. Among them, researchers found that *NANOG*-expressing human bone marrow mesenchymal stem cells (MSCs) had much higher capabilities for expansion and osteogenesis [40]. A GWAS analysis in humans has reported that *COX7A2* and *RIMS1* were identified as candidate genes for osteoarthritis [41,42], and *COX7A2* was highly expressed in cartilage [43]. In another study, *CACNG1* is thought to participate in modulating postmenopausal osteoporosis via Ca^2+^ regulation during muscle contraction [44]. Go analysis results showed that *CACNG1* is involved in the process of calcium channel regulator activity. In recent years, the forkhead box protein (FOX) family has been reported to play important roles in different kinds of physiological bone processes [45]. For example, during the process of osteogenic differentiation, the expression of *FOXP1* was enhanced under the regulation of circ*FOXP1* [46]. Additionally, a study also indicated that a decrease in *FOXP1* level was accompanied by the augmented expression of PPARγ, which serves as a master regulator in adipocyte differentiation [47]. Over-expression of the *FOXP1* in adipose cells affects adaptive thermogenesis and boosts diet-induced obesity [48]. GO results showed that *FOXP1* is involved in the process of osteoclast development. It has been established that fat is the primary factor impairing body size. *FOXP1* might play essential roles in lipid metabolism and osteogenic differentiation, so the *FOXP1* gene could be considered as a strong candidate gene for the CC. In sheep, *BMPR1B* was detected as a candidate gene for variation in mature size [49]. In addition, *BMPR1B* participates in skeletal patterning, and is mainly found in differentiated chondrocytes and osteoblasts, and is also expressed in mesenchymal pre-cartilage condensations [50,51]. *COX6C* was identified as a critical gene involved in regulating osteoblast mineralization and mitochondrial bioenergetics [52].

TNB and NSB are key indicators for measuring sow reproduction performance. In TNB, gene annotation results showed that *SMG1* was the closest gene for SNPs CASI0010189 on chromosome 3. It has been reported that *SMG1* is a necessary kinase for mouse embryogenesis. The absence of *SMG1* would be fatal to embryonic development, mainly due to the profound developmental defects (e.g., brain and heart) [53]. In daily production, high proportions of NSB cause serious economic losses. We here found several candidate genes, including *ARAP2*, *GTF2H5*, *SLC29A1*, *FAM49B*, *CNTNAP4*, *KIT*, *POR*, *SPAM1*, *TFAP2B*, *BPIFA3*, and *NEDD9,* that were potentially associated with NSB. For *ARAP2*, a GWAS analysis in Chinese Holstein cows found that it was associated with loin strength (LS), a reproduction-associated body-shape trait [25]. Cows with a weak loin usually possess a sinking uterus, which could easily lead to reproductive system diseases [54]. Whole exome sequencing for a male neonate with premature rupture of membranes and intrauterine growth restriction showed that *GTF2H5* gene mutations induced severe clinical manifestations including multiple-organ failure [55]. *SLC29A1*, also called *ENT1*, belongs to the solute carrier (SLC) family. It was initially found from a human placental DNA library [56]. According to previous studies, the change in SLC drug transporter expression impacts the drug’s disposition and pharmacokinetics in the placenta, and a variety of substrates including toxins, nutrients, and signaling molecules are transported by SLC transporters during pregnancy [57,58]. Although previous research has not provided evidence to support a causal association between them, *SLC29A1* might be a potential causal biomolecule for NSB. *CNTNAP4* is a member of the neurexin superfamily and is essential for synaptic function and neural development [59]. In addition, a study showed that *CNTNAP4* is necessary for the proliferation of embryonic neural progenitor cells (NPCs) in mice [60]. Notably, a GWAS study in beef cattle found that the *CNTNAP4* gene was associated with fertility traits [61]. In the *KIT* mutation murine model, a reduction in placental vascularization was present, which finally resulted in the placenta developing irregularly and embryos presenting with severe growth retardation and dying in the uterus [62,63]. Moreover, GO analysis found that the *KIT* gene participates in the glycosphingolipid metabolic process. Intriguingly, an earlier study indicated that glycosphingolipid composition is important for the growth and differentiation of trophoblast cells during pregnancy [64]. These results suggested that *KIT* might be the important regulator for healthy embryonic development. It should be regarded as a strong candidate gene for NSB.

Recently, a study of prenatal and postnatal deaths for three sibling fetuses revealed that the combination of CNV and SNV in the mutated *POR* gene was responsible for their death [65]. *SPAM1*, a hyaluronidase, played roles in binding to sperm during capacitation, elevating cumulus dispersal efficiency and the capacity of sperm to across the cumulus of oocytes [66,67]. KEGG results showed that *SPAM1* is involved in glycosaminoglycan degradation pathways. Of note, in the developmental process of embryos and fetuses, glycosaminoglycans play essential roles in cell morphogenesis, growth, differentiation, and cell migration [68,69]. *SPAM1* may affect the expression pattern of the glycosaminoglycan degradation pathway, which ultimately affects embryonic development. *TFAP2B* is a member of the transcription factor *TFAP2* family, also called *AP-2*. Earlier researchers found that *TFAP2A*-deficient mice suffered from anomalies, such as anencephaly, congenital heart disease, and body-wall defects [70,71,72]. According to earlier reports, *TFAP2B*-deletion led to neonatal mice death due to the massively increased apoptotic cell death for renal epithelial cells [73,74]. A subsequent study demonstrated that *TFAP2B*^−/−^ mice exhibited developmental defects in the ductus arteriosus and limbs [75]. Thus, *TFAP2B* could be regarded as a strong candidate gene for NSB, because of its central role in teratogenicity. *NEDD9*, a target gene of TGF-beta, was identified in Yorkshire sows as a candidate core gene for litter size [76]. Besides, *NEDD9* regulates the invasion and proliferation of ectopic endometriotic stromal cells [77].

The duration of pigs’ farrowing and estrous cycles are considered as critical factors of reproduction traits. A short duration of farrowing is vital for piglet survival as a delay can elevate the number of stillborn [78]. Another research indicated that the estrous cycles might affect litter size by influencing the ovulation rate [79]. Besides, the reproduction records in multiparous sows are also important phenotypes for the GWAS study. However, owing to the effects of the African Swine fever virus, during that period, many sows’ mating was suspended in the Shaziling pig farm, so that limited information for reproduction traits were obtained from that Shaziling pig farm, and only the first-parity records were included in this GWAS study. In the future, these reproduction traits-related factors shall be included in the model and the results shall be verified in a larger Shaziling population. In addition, metabolism and metabolic biomarkers have been previously implicated in studies of sows’ fertility, and the metabolite biomarkers could be the important phenotype in terms of sows’ reproduction potential [80]. Moreover, a recent report in our group for comparisons of carcass traits, meat quality, and serum metabolome between Shaziling and Yorkshire pigs has suggested that a higher serum L-carnitine content is a promising indicator for better meat quality of pigs [12]. Therefore, serum metabolite should also be regarded as a key trait for GWAS research in future study.

Chip-based SNP markers have been widely used in genomic studies because of their abundance in the genome and their low cost [81]. In this study, we applied the Porcine SNP 50k Chip for genotyping samples. In the early phase of the project for Shaziling pigs, the main object of genotyping samples was to identify the pureblood individuals and pedigree relationships for the Shaziling population. Afterwards, based on the previous results, we further measured the body size and reproduction traits for the pure Shaziling pigs. Then, this first GWAS study for the Shaziling population was carried out. With the development of sequencing technology and an analysis method, genotyping-by sequencing (GBS) is also used for identifying variants nowadays because of its lower cost per data point (especially for low-pass sequencing data) and the avoidance of ascertainment bias during genotyping [81]. Recent studies have found that GWAS using low-pass sequencing data showed similar results to those with SNP chip data, but may require much larger sample sizes to show measurable advantages [82]. Compared with other genotyping methods like GBS, although the SNP chip data in this study may provide limited information throughout the whole genome, it provided preliminary and useful information for the future selection and breeding of Shaziling pigs. Meanwhile, we shall acknowledge that a platform such as GBS should also be used for the detection of candidate functional SNP markers in Shaziling pigs. In this study, we detected some genetic markers that may affect the body measurement and reproduction traits of the Shaziling population. Information on the genomic regions explored in this current work can accelerate the identification of candidate genes for measurement and reproduction phenotypes. However, we did not identify any significant SNPs for NHB and MUMM. This may be partly due to the inadequate sample size used in the current study. Eventually, these outcomes provide meaningful information for genomic selection in the Shaziling pigs for high-efficiency genetic improvement.

## 5. Conclusions

By the genome-wide association study in the Shaziling pig breed, we identified 31 SNPs in total that were potentially associated with body size and reproductive traits of interest. Eighteen functional genes, including *GLP1R*, *NFYA*, *NANOG*, *COX7A2*, *CACNG1*, *FOXP1*, *BMPR1B*, *COX6C*. *SMG1*, *ARAP2*, *GTF2H5*, *SLC29A1*, *CNTNAP4*, *KIT*, *POR*, *SPAM1*, *TFAP2B*, and *NEDD9,* were identified as important candidate genes that may regulate the underlying genetic architecture of porcine body size and reproductive traits. Although the current study is limited by its small sample size, it helps us to understand the genetic basis of porcine body measurement and reproduction traits and could be potentially applied in pig breeding programs.

## Figures and Tables

**Figure 1 genes-14-00522-f001:**
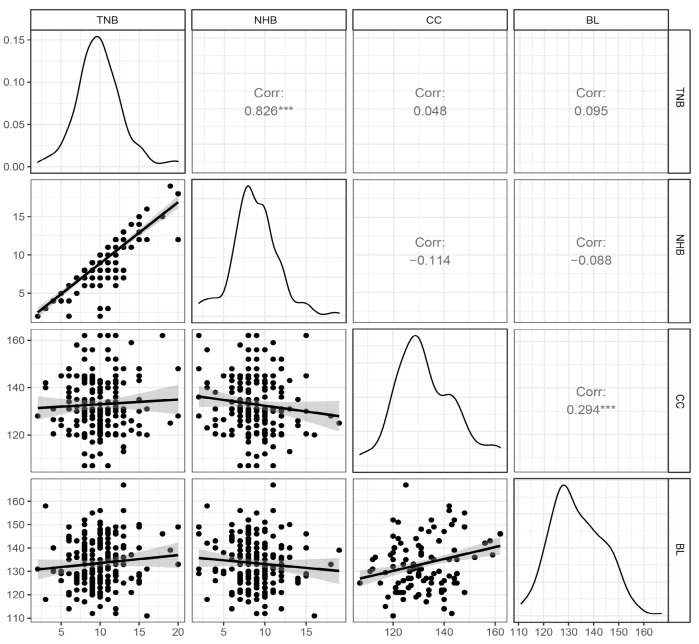
The normal distribution and correlation analysis between body size and reproduction traits. “***” represents for *p* < 0.001. TNB: total number born. NHB: number of healthy births. CC: chest circumstance. BL: body length.

**Figure 2 genes-14-00522-f002:**
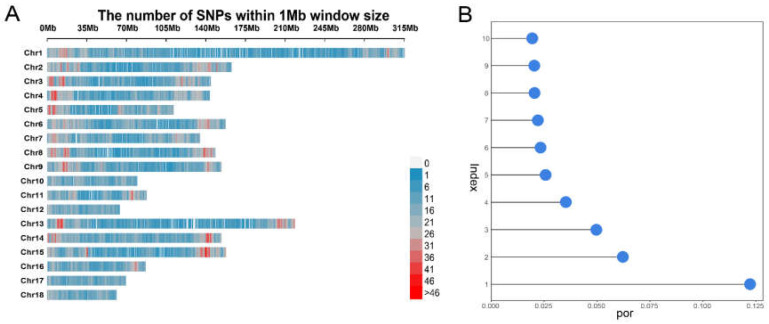
The distribution of SNPs on autosomal chromosomes and the principal component analysis. (**A**), The distribution of SNPs within a 1 Mb window size. (**B**), The lollipop-plot of top ten principal components. Index means top ten principal components and por represents proportion of explained variance.

**Figure 3 genes-14-00522-f003:**
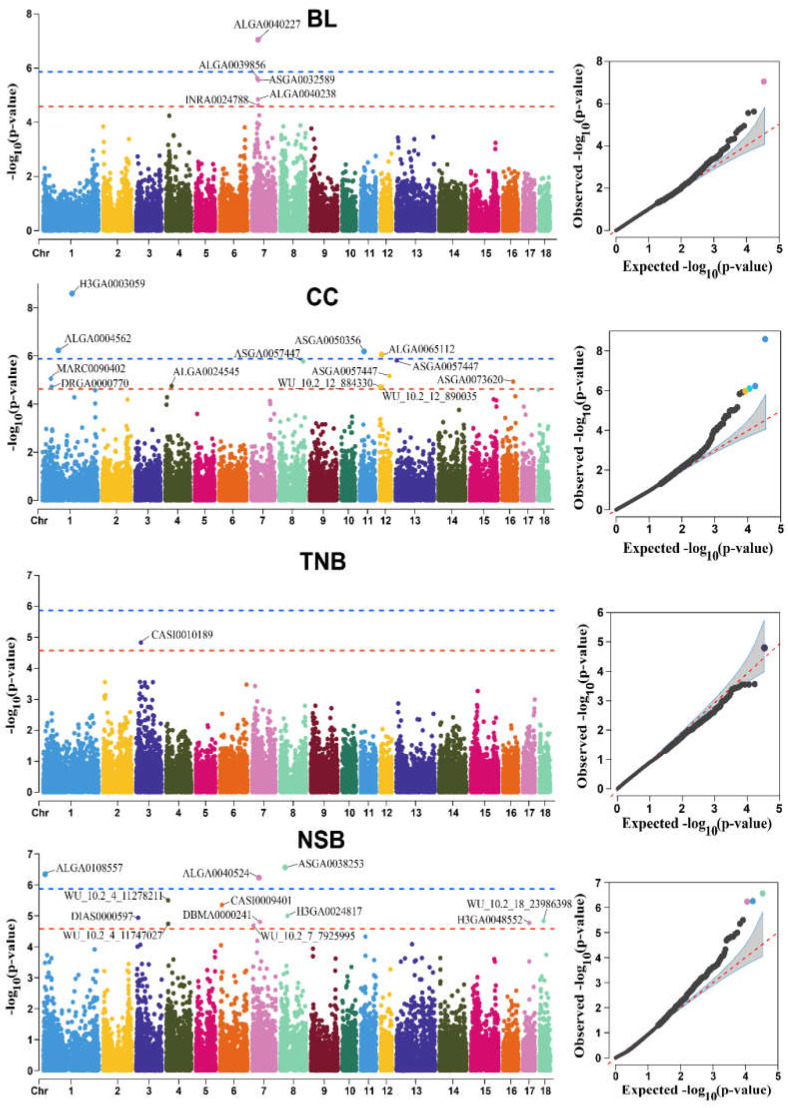
Manhattan and quantile-quantile (QQ) plots of the observed −log_10_(*p*-values) for BL, CC, TNB, and NSB in Shaziling pigs. The horizontal red and blue dashed lines in the Manhattan plots indicate the suggestive level (2.38 × 10^−5^) and significant level (1.19 × 10^−6^), respectively. The QQ plots show the observed −log_10_-transformed *P*-values (y-axis) and the expected −log_10_-transformed *p*-values (x-axis).

**Figure 4 genes-14-00522-f004:**
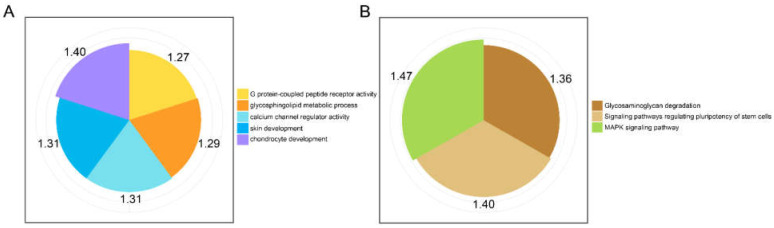
GO annotation and KEGG pathway analysis for candidate genes. (**A**), biological process of GO terms. (**B**), KEGG pathways. The numbers in the outermost layer indicated the −log_10_ adjusted *p*-values.

**Table 1 genes-14-00522-t001:** Summary statistics of body measurement and reproduction traits in the Shaziling population.

Population	Trait ^a^	N	Mean	SD	Minimum	Maximum	h^2^	CV (%)
Shaziling	BL (cm)	190	133.2	10.6	110	158	0.15	7.96%
	CC (cm)	190	132.9	12.0	107	162	0.14	9.03%
	TNB	190	9.88	2.90	2	20	<0.01	-
	NHB	190	8.81	2.81	2	19	<0.01	-
	NSB	190	0.14	0.48	0	2	<0.01	-
	MUMM	190	0.93	1.60	0	8	0.02	-

^a^: BL = body length, CC = chest circumference, TNB = total number born, NHB = number of healthy births, NSB = number of stillborn, MUMM: number of mummified pigs.

**Table 2 genes-14-00522-t002:** Potential single-nucleotide polymorphisms (SNPs) and candidate genes identified by the genome-wide association study for body size and reproduction traits.

Trait	SSC ^1^	SNP	Position	*p*-Value	MAF ^2^	β ^3^	Distance (bp)	Candidate Genes
BL	7	ALGA0040227	34835986	9.01 × 10^−8^	0.41 (A/G)	−6.49	301630	*GLP1R*
BL	7	ALGA0039856	30719424	2.36 × 10^−6^	0.15 (A/G)	−7.27	within	*SNRPC*
BL	7	ASGA0032589	36378825	2.76 × 10^−6^	0.22 (A/G)	−6.24	7890	*NFYA*
BL	7	ALGA0040238	34856640	1.42 × 10^−5^	0.33 (G/A)	−5.48	322284	*GLP1R*
BL	7	INRA0024788	34978383	2.38 × 10^−5^	0.35 (A/G)	−5.31	444027	*GLP1R*
CC	1	H3GA0003059	168650104	2.56 × 10^−9^	0.02 (A/G)	−4.69	414030	*NANOG*
CC	1	ALGA0004562	90654890	5.95 × 10^−7^	0.06 (G/A)	−2.54	41856	*COX7A2*
CC	11	ASGA0050356	24088736	7.95 × 10^−7^	0.02 (A/G)	−3.41	224869	*FAM216B*
CC	12	ALGA0065112	13563645	1.05 × 10^−6^	0.01 (A/C)	−4.85	89321	*CACNG1*
CC	13	ASGA0057447	52893776	1.24 × 10^−6^	0.01 (T/A)	−5.19	16884	*FOXP1*
CC	8	ALGA0111294	124178084	1.47 × 10^−6^	0.04 (G/A)	−2.56	360686	*BMPR1B*
CC	12	MARC0074335	56183179	6.87 × 10^−6^	0.02 (A/G)	−3.11	within	*DNAH9*
CC	1	MARC0090402	48528854	8.86 × 10^−6^	0.05 (A/G)	−2.66	175464	*ENSSSCG00000060802*
CC	16	ASGA0073620	60469163	1.08 × 10^−5^	0.01 (G/A)	−3.85	112426	*CCNG1*
CC	4	ALGA0024545	37210968	1.78 × 10^−5^	0.05 (G/A)	−2.58	121604	*COX6C*
CC	1	DRGA0000770	51978066	1.90 × 10^−5^	0.03 (G/A)	−2.34	within	*RIMS1*
CC	12	WU_10.2_12_884330	884330	1.91 × 10^−5^	0.04 (G/A)	−2.22	within	*CCDC57*
CC	12	WU_10.2_12_890035	890035	1.91 × 10^−5^	0.04 (A/G)	−2.22	within	*CCDC57*
TNB	3	CASI0010189	26899862	1.60 × 10^−5^	0.03 (A/C)	4.26	129540	*SMG1*
NSB	8	ASGA0038253	27689768	2.75 × 10^−7^	0.02 (G/A)	0.88	81655	*ARAP2*
NSB	1	ALGA0108557	8855372	5.59 × 10^−7^	0.02 (A/C)	1.08	61868	*GTF2H5*
NSB	7	ALGA0040524	39239703	5.87 × 10^−7^	0.01 (G/A)	1.25	5018	*SLC29A1*
NSB	4	WU_10.2_4_11278211	11278211	3.18 × 10^−6^	0.01 (A/G)	0.90	418666	*FAM49B*
NSB	6	CASI0009401	10987986	4.47 × 10^−6^	0.03 (A/G)	0.68	329863	*CNTNAP4*
NSB	8	H3GA0024817	41595542	1.01 × 10^−5^	0.07 (A/G)	0.45	103236	*KIT*
NSB	3	DIAS0000597	10273754	1.16 × 10^−5^	0.02 (G/A)	0.88	2896	*POR*
NSB	18	WU_10.2_18_23986398	23986398	1.46 × 10^−5^	0.08 (A/G)	0.39	396451	*SPAM1*
NSB	7	DBMA0000241	45173179	1.58 × 10^−5^	0.09 (G/A)	0.44	402885	*TFAP2B*
NSB	17	H3GA0048552	36746539	1.70 × 10^−5^	0.15 (A/G)	0.31	within	*BPIFA3*
NSB	4	WU_10.2_4_11747027	11747027	1.81 × 10^−5^	0.01 (C/A)	0.96	45942	*ENSSSCG00000060440*
NSB	7	WU_10.2_7_7925995	7925995	2.07 × 10^−5^	0.12 (A/C)	0.34	11934	*NEDD9*

BL, body length; CC, chest circumference; TNB, total number born; NSB, number of stillborn. ^1^ Sus scrofa chromosome. ^2^ allele frequency of first listed marker. ^3^ allele substitution effect.

## Data Availability

The data presented in this study are available on request from the corresponding author.

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
