# Peer review of "Genome-Wide Association Analysis Identified Variants Associated with Body Measurement and Reproduction Traits in Shaziling Pigs"

_genes, 2023, doi:10.3390/genes14020522_

Round 1
Reviewer 1 Report
Summary:
In this study, the investigators record two body measurement traits and four reproduction traits in first parity from the 190 Shaziling sows and perform a genome-wide association study (GWAS) between the SNPs and the six phenotypes. Although, based on the results, the correlation between body size and reproduction phenotypes was not statistically significant, the paper is somewhat helpful for identifying the molecular markers for pig breeding.
1. The title "Identifying the complex genetic architecture of body measurement and reproduction traits in Shaziling pigs" are exaggerated. The paper discussed two body measurement traits and four reproduction phenotypes and performed GWAS. Therefore, I would suggest modifying the title.
2. Body length is usually associated with growth traits, meat production, etc. The paper did not discuss anything related to this. There are few references supporting the statement, "Previous study has indicated that body measurement traits could be the indicator traits for reproduction performance." The authors cited a recent publication ref 20 (Lu et al., 2021), which is the study of dairy cows. Dairy cows are different from pigs. The current reference does not fully support the idea that body measurement traits are indicators of reproduction performance. Please provide more references to support this statement.
3. The authors discuss the limitations of having an inadequate sample size (190 sows) used in the current study, which is great. Some traits, for example, serum metabolome, are usually key reproduction traits. Please explain the reason for not including this kind of data or discuss the limitation.
4. The duration of pigs' farrowing and estrous cycles are considered critical factors of reproduction traits. Therefore, please provide the data on the duration of the pigs' farrowing and estrous cycle or discuss why not include these parts of the data.
5. The authors genotype samples by using SNP chips, and please discuss the reason why not using the sequencing method. Alternatively, discuss the limitations in the discussion part.
6. The litter size in multiparous sows is a better factor. In this paper, the litter records were collected in the first parity. Please provide data on the second or third parity or discuss why choosing to collect litter records in the first parity only.
Author Response
In this study, the investigators record two body measurement traits and four reproduction traits in first parity from the 190 Shaziling sows and perform a genome-wide association study (GWAS) between the SNPs and the six phenotypes. Although, based on the results, the correlation between body size and reproduction phenotypes was not statistically significant, the paper is somewhat helpful for identifying the molecular markers for pig breeding.
Comment 1-1: The title "Identifying the complex genetic architecture of body measurement and reproduction traits in Shaziling pigs" are exaggerated. The paper discussed two body measurement traits and four reproduction phenotypes and performed GWAS. Therefore, I would suggest modifying the title.
Response 1-1: Thank you very much for pointing out this issue. We agree with your suggestion. In this revised manuscript, the title has been modified to “Genome-wide association analysis identified variants associated with body measurement and reproduction traits in Shaziling pigs”, to make it more suitable for this research.
Comment 1-2: Body length is usually associated with growth traits, meat production, etc. The paper did not discuss anything related to this. There are few references supporting the statement, "Previous study has indicated that body measurement traits could be the indicator traits for reproduction performance." The authors cited a recent publication ref 20 (Lu et al., 2021), which is the study of dairy cows. Dairy cows are different from pigs. The current reference does not fully support the idea that body measurement traits are indicators of reproduction performance. Please provide more references to support this statement.
Response 1-2: Thank you very much for your careful review and good suggestion. In our previous manuscript, we had mentioned two related references regarding the body length’s associations with body weight and meat production, but had not summarize the relations between body measurement traits and growth and meat production. Here, based your suggestions, in the revised version, we have added the summary sentence and added another related reference (lines 257-261, reference 33). As for the statement, "Previous study has indicated that body measurement traits could be the indicator traits for reproduction performance", we have supplemented other three references (references 22-24) in pigs to provide more evidences on that statement (lines 240-241).
Comment 1-3: The authors discuss the limitations of having an inadequate sample size (190 sows) used in the current study, which is great. Some traits, for example, serum metabolome, are usually key reproduction traits. Please explain the reason for not including this kind of data or discuss the limitation.
Response 1-3: Thank you for your advice. Based your suggestions, in limitation part of discussion, we have added the discussion for the importance of serum metabolome in indicating reproduction potential of pigs in the further study (lines 366-374).
Comment 1-4: The duration of pigs' farrowing and estrous cycles are considered critical factors of reproduction traits. Therefore, please provide the data on the duration of the pigs' farrowing and estrous cycle or discuss why not include these parts of the data.
Response 1-4: Thank you for pointing this out. In our study, we haven’t used the data of duration of pigs' farrowing and estrous cycles in the GWAS study because the data were collected in early stage. However, owing to the effects of African Swine fever virus, during that period in 2019, many sows’ mating was suspended in the Shaziling pig farm, so that only insufficient information for reproduction traits were obtained from that Shaziling pig farm. It is difficult to achieve the comprehensive data for estrous cycles and duration of farrowing, some were missing or incorrect. Therefore, only limited reproduction traits in the first-parity were used for this GWAS study. It was nice of you to remind us the importance of pigs' farrowing time and estrous cycles in the GWAS study. We have supplemented the related discussion for the importance of duration of pigs' farrowing and estrous cycles in lines 356-366. In the future GWAS study related with reproduction performance of Shaziling pigs, these reproduction traits-related factors shall be included in the model. Thanks again for your scientific guidance.
Comment 1-5: The authors genotype samples by using SNP chips, and please discuss the reason why not using the sequencing method. Alternatively, discuss the limitations in the discussion part.
Response 1-5: Thank you for this valuable point. As displayed in lines 374-391, according your suggestions, we have added the reason why we used Porcine 50K SNP Chip and discussions for the limitations of this study.
Comment 1-6: The litter size in multiparous sows is a better factor. In this paper, the litter records were collected in the first parity. Please provide data on the second or third parity or discuss why choosing to collect litter records in the first parity only.
Response 1-6: Thanks for this question and suggestion. Like what we describe in response 1-4, owing to the effects of African Swine fever virus, during that period in 2019, many sows’ mating was suspended in the Shaziling pig farm, some work for recording were suspended or incorrect, which might result in the occurrence of long weaning time, and some sows cannot even conceive naturally. It is hard to achieve the comprehensive data for the litter size records in multiparous sows. Thus, only the first parity records were used to perform GWAS study. We have added the related descriptions in lines 359-366.

Reviewer 2 Report
The article reports an interesting topic, and of great interest for livestock production, however, the methods need to be better explained, please see the comments below:
Page 2- L53 – add “be” between might and important
Page 3, L105- delete % or use 5%
Page 4, L157 to 159 – you have to explain better the methods used for the threshold; you are using Bonferroni correction ( 0.05/41857), so it isn`t a suggestive threshold, but I don`t know what is 2.38E-05 (1/41,857) you have to describe the methods, based in what methods or theory you suggested this threshold, I must be explained
Page 5, 174 - I did not understand why you included the first four principal components in the GWAS model as covariates to correct for population stratification, you have female sows from the same breed raised in the core conservation farm, how is it possible stratification? if so, you have to explain better de animals’ background. where is the PCA chart?
Author Response
The article reports an interesting topic, and of great interest for livestock production, however, the methods need to be better explained, please see the comments below:
Comment 1-1: Page 2- L53 – add “be” between might and important.
Respond 1-1: Thanks for the careful review. We have added it in line 56.
Comment 1-2: Page 3, L105- delete % or use 5%.
Respond 1-2: We have modified it into 5% (line 107). Thanks.
Comment 1-3: Page 4, L157 to 159 – you have to explain better the methods used for the threshold; you are using Bonferroni correction (0.05/41857), so it isn`t a suggestive threshold, but I don`t know what is 2.38E-05 (1/41,857) you have to describe the methods, based in what methods or theory you suggested this threshold, I must be explained
Respond 1-3: Thank you for pointing this out. We apologize for our description mistake. Now, we have rewritten this sentence. The right sentence is gonging like “the genome-wide significant and suggestive threshold of the Shaziling population were 1.19E-06 (0.05/41,857) and 2.38E-05 (1/41,857), respectively”, which can be found in lines 163-164. According the other researches (PMID: 36011365, PMID: 36506319), based on the number of filtered SNPs, the Bonferroni-adjusted genome-wide significant threshold was 0.05/N, and the suggestive threshold was 1/N, N is the total number of filtered SNPs. We have added the description for the method in the Materials and Methods section (lines 136-139). Thanks for your careful examination and consideration.
Comment 1-4: Page 5, 174 - I did not understand why you included the first four principal components in the GWAS model as covariates to correct for population stratification, you have female sows from the same breed raised in the core conservation farm, how is it possible stratification? if so, you have to explain better de animals’ background. where is the PCA chart?
Respond 1-4: Thank you for these two valuable points. In the previous manuscript, owing to our negligence, we added word “core” into the sentence which describe the information about the pig farm, actually, it was a conservation farm. Thus, we have deleted the word “core” line 83. Regarding the background of this Shaziling conservation farm, some of the pigs are purchased from peasant households because of the influence of African swine fever (ASF). We added the impacts of ASF in lines 360-364. We also added the background of this research lines 376-380. “Actually, in the current study, in the early phase of the project for Shaziling pigs, the main object of genotyping samples was to identify the pureblood individuals and pedigree relationships for the Shaziling population. Afterwards, based on the previous results, we further measured the body size and reproduction traits for the pure Shaziling pigs. Then, this first GWAS study for Shaziling population was carried out.” Therefore, it is reasonable that the subjects used for GWAS may have population stratification. As for the PCA chart, we have provided the related description in the main text, which can be found in lines 179 and 409, and put it in the supplementary materials (Figure. S1). According to the result of PCA chart, we found that PC5 and PC6 can’t disperse Shaziling population, we thus selected the first four principal component into GWAS model. Thanks again for the time and great efforts you’ve spent on our manuscript. We're very grateful to you.
Round 2
Reviewer 1 Report
Thanks very much for the responses. Please double-check the language before publication.